# The Associations of Iron Related Biomarkers with Risk, Clinical Severity and Mortality in SARS-CoV-2 Patients: A Meta-Analysis

**DOI:** 10.3390/nu14163406

**Published:** 2022-08-19

**Authors:** Shuya Zhou, Huihui Li, Shiru Li

**Affiliations:** Department of Epidemiology and Health Statistics, School of Public Health, Qingdao University, Qingdao 266071, China

**Keywords:** iron, ferritin, hemoglobin, TIBC, COVID-19

## Abstract

The coronavirus disease 2019 (COVID-19), caused by severe acute respiratory syndrome coronavirus 2 (SARS-CoV-2), is spreading rapidly around the world and has led to millions of infections and deaths. Growing evidence indicates that iron metabolism is associated with COVID-19 progression, and iron-related biomarkers have great potential for detecting these diseases. However, the results of previous studies are conflicting, and there is not consistent numerical magnitude relationship between those biomarkers and COVID-19. Thereby, we aimed to integrate the results of current studies and to further explore their relationships through a meta-analysis. We searched peer-reviewed literature in PubMed, Scopus and Web of Science up to 31 May 2022. A random effects model was used for pooling standard mean difference (SMD) and the calculation of the corresponding 95% confidence interval (CI). *I*^2^ was used to evaluate heterogeneity among studies. A total of 72 eligible articles were included in the meta-analysis. It was found that the ferritin levels of patients increased with the severity of the disease, whereas their serum iron levels and hemoglobin levels showed opposite trends. In addition, non-survivors had higher ferritin levels (SMD (95%CI): 1.121 (0.854, 1.388); Z = 8.22 *p* for Z < 0.001; *I*^2^ = 95.7%, *p* for *I*^2^ < 0.001), lower serum iron levels (SMD (95%CI): −0.483 (−0.597, −0.368), Z = 8.27, *p* for Z < 0.001; *I*^2^ = 0.9%, *p* for *I*^2^ =0.423) and significantly lower TIBC levels (SMD (95%CI): −0.612 (−0.900, −0.324), Z = 4.16, *p* for Z < 0.001; *I*^2^ = 71%, *p* for *I*^2^ = 0.016) than survivors. This meta-analysis demonstrates that ferritin, serum iron, hemoglobin and total iron banding capacity (TIBC) levels are strongly associated with the risk, severity and mortality of COVID-19, providing strong evidence for their potential in predicting disease occurrence and progression.

## 1. Introduction

Coronavirus disease 2019 (COVID-19), caused by severe acute respiratory syndrome coronavirus 2 (SARS-CoV-2) [1], has become a global pandemic, resulting in 525,646,754 confirmed cases and 6,299,346 deaths as of 31 May 2020 [2,3]. The rapid spread of this disease has put enormous pressure on local medical institutions and their finances. Thus, it is very critical to identify and prevent the spread of COVID-19 early.

The clinical manifestations of COVID-19 are various, from asymptomatic infection to death. Therefore, it is very necessary to find a reliable early biomarker to identify the emergence and progression of disease [3,4]. Numerous studies indicated that iron distribution was closely linked to the onset and progression of COVID-19. In the process of viral pathogen invasion, the defense system is very crucial [5], especially the immune system, which relies on the supply of micronutrients. Iron is not only an important component of micronutrients, but also plays an important role in various fundamental biological processes between human and pathogen, ranging from deoxyribonucleic acid (DNA) synthesis to adenosine triphosphate (ATP) generation [6,7]. Furthermore, some symptoms of COVID-19, such as pneumonia, thrombo-embolism and acute respiratory distress syndrome (ARDS), are also related to iron by its functions in the immune system and circulatory system. Therefore, the application of iron-related biomarkers in identifying infection and severity in COVID-19 patients has attracted widespread attention.

Though serum or plasma iron is a main indicator for iron homeostasis, ferritin, hemoglobin, hepcidin, TIBC and transferrin saturation (TSAT) would explain iron distribution more roundly. Previous studies have investigated the relationships among iron, ferritin, hemoglobin, hepcidin, TIBC, TSAT and COVID-19, but their results are conflicting and inconsistent. Most studies about serum iron have shown that lower iron levels are found in more at-risk groups, such as non-survivors compared to survivors [8,9,10,11,12,13], severe disease groups compared to non-severe disease groups [10,13,14,15,16] and cases to controls [10,16,17,18,19,20,21], whereas others expressed the opposite results [7,22,23]. In all relevant studies, the hemoglobin levels were found to be lower in severe groups than those in controls [7,10,24] or non-severe groups [22,25,26,27,28,29]. However, in the comparisons between severity [7,10,15,17,18,19,23,24,30,31,32,33,34,35,36,37] and mortality [10,30,38,39,40,41], there was not a consistent numerical magnitude relationship between them. In addition, studies on TIBC [8,10,12,13] and TSAT [8,13] only refer to mortality. Most of those studies found lower levels in non-survivor groups, except for one about TSAT [10]. Lower levels of hepcidin in cases than controls was documented in a study [10], whereas the other studies reported an inverse result [18,20,24]. Unusually, more dangerous groups had higher ferritin levels in almost all relevant comparisons [4,6,10,11,12,13,14,15,17,18,19,20,21,22,24,25,26,27,28,29,30,32,34,35,36,37,38,39,40,42,43,44,45,46,47,48,49,50,51,52,53,54,55,56,57,58,59,60,61,62,63,64,65,66,67,68,69,70,71,72].

Considering that any individual study may not have the sufficient power to obtain a reliable conclusion, this meta-analysis was conducted to: (1) Summarize and evaluate the results of numerous papers on the relationships between serum iron, ferritin, hemoglobin, TIBC, TSAT and hepcidin levels and the mortality and clinical severity of SARS-CoV-2 patients. (2) Assess the potential between-study heterogeneity; and eventually investigate the potential publication bias.

## 2. Materials and Methods

### 2.1. Literature Search and Data Selection Criteria

Our meta-analysis searched peer-reviewed literature on PubMed [73], Web of Science [74] and Scopus [75] up to May 2022 published in English. Our keyword combinations included (Fe OR iron OR ferritin) AND (COVID-19 OR human coronavirus disease 2019 OR SARS-CoV-2 OR severe acute respiratory syndrome coronavirus 2). Moreover, in order to identify extra studies which were not included by databases, we reviewed the references of all searched literature.

### 2.2. Data Selection Criteria

Articles were included based on the following four criteria: (1) the cases had COVID-19; (2) observational study design; (3) samples were from blood; (4) data were reported as mean ± standard deviation (SD) or the other forms could be translated to mean ± SD. Some studies were excluded if they were: (1) experimental studies (2) reviews.

If the data were duplicated in one study, only the one that involved the largest number of cases was included. Two investigators searched for the articles and extracted data independently. The disagreements about eligibility of an article were settled by discussion.

### 2.3. Data Extraction

On each article, the following information and data were recorded: the first author, county (continent), publication year, study design, disease assessment, outcome groups, gender, mean age, number of cases, number of controls, number of non-survivors, number of survivors, numbers of severe and non-severe cases, mean ± SD of these groups and data units. If just standard error mean (SEM) was available, SD was calculated by the formula SEM = SD/$\sqrt{n}$. If there were available median and interquartile range, mean and SD were to be calculated using the website [76].

### 2.4. Statistical Analyses

The meta-analysis was performed using the software Stata 15.0 (Stata Corporation, College Station, TX, USA). In order to estimate the association between serum-iron-related biomarkers and the onset and progression of COVID-19, we used the standard mean difference (SMD) with the corresponding 95% confidence interval (CI) as the effect size. The SMD is the ratio of the mean difference to the pooled standard deviation. *I*^2^ was used to assess the heterogeneity among studies. For the purpose of pooling effect sizes and all the analyses, we used a random effects model. All reported probabilities (*p*-values) were two-sided, and *p* < 0.05 was recognized as statistically significant. If meta-regression demonstrated sources of heterogeneity in continent, publication year, study design and subgroup analyses were performed.

## 3. Results

### 3.1. Characteristics of Studies

After the preliminary search, 6916 articles were identified, 2040 from PubMed, 1390 from Web of Science and 3486 from Scopus (Figure 1). After screening the duplicated and irrelative articles, 157 articles were included. Among them, 85 articles were further excluded because: six were mechanism research; the data of 16 were not analyzable; 25 expressed the data in odds ratios, relative risks or correlation coefficients; 16 were systematic reviews; and 22 were experimental studies. Eventually, a total of 72 eligible articles were involved in this meta-analysis: 17 were case–control studies, 20 studies were cohort studies and 35 were cross-sectional studies. (Table 1)

### 3.2. Ferritin Level and COVID-19

A total of twenty-nine studies assessed the connections between ferritin levels and the mortality of SARS-CoV-2 patients in this meta-analysis, involving 2131 non-survivors and 7813 survivors. The ferritin levels were significantly higher in the dead patients than that in survivors (SMD (95%CI): 1.121 (0.854, 1.388); Z = 8.22 *p* for Z < 0.001; *I*^2^ = 95.7%, *p* for *I*^2^ < 0.001; Table 2, Figure 2). A subgroup analysis by publication year indicated that the dead had higher ferritin levels in 2020 (SMD (95%CI): 1.881 (1.137, 2.625); Appendix A), 2021 (SMD (95%CI): 0.847 (0.575, 1.119)), and 2022 (SMD (95%CI): 0.550 (0.393, 0.707); Appendix A). More details about subgroup analysis are in Table 2.

Seventeen studies about ferritin were included in the COVID-19 risk analysis, including 964 cases and 966 controls. The overall comparison demonstrated that cases had higher ferritin levels than controls (SMD (95%CI): 1.383 (0.792, 1.975); Z = 4.58 *p* for Z < 0.001; *I*^2^ = 96.3%, *p* for *I*^2^ < 0.001; Table 3, Figure 3). In the subgroup analysis by study type, higher ferritin levels were found in cases compared with controls about risk analysis (SMD (95%CI): 0.872 (0.443, 1.300); Appendix A). More detailed records of subgroup analysis are shown in Table 3.

In terms of the relationship between ferritin and the severity of COVID-19, a total of thirty-nine studies were included, involving 1681 severe, 1337 non-severe, 433 moderate and 1416 mild individuals. The results reported that the ferritin levels of severe groups were higher than those of the non-severe groups (SMD (95%CI): 0.864 (0.389, 1.338); Z = 3.57 *p* for Z < 0.001; *I*^2^ = 96.00%, *p* for *I*^2^ < 0.001; Table 3, Appendix A) and mild groups (SMD (95%CI): 1.414 (0.995, 1.834); Z = 6.61 *p* for Z < 0.001; *I*^2^ = 92.7%, *p* for *I*^2^ < 0.001; Table 3, Appendix A). In addition, a significant difference was also observed between moderate groups and mild groups (SMD (95%CI): 1.551 (0.535, 2.566); Z = 2.99 *p* for Z = 0.003; *I*^2^ = 97.2%, *p* for *I*^2^ < 0.001; Table 3, Appendix A). The subgroup analysis based on publication year indicated that similar association was found in 2020 (SMD (95%CI): 2.802 (0.678, 4.925); Appendix A) and in 2021(SMD (95%CI): 0.400 (0.207, 0.592); Appendix A). With regard to continent, significant differences were found among Asia (SMD (95%CI): 0.976 (0.362, 1.591); Appendix A), Europe (SMD (95%CI): 0.581 (0.259, 0.903); Appendix A) and Africa (SMD (95%CI): 5.319 (4.718, 5.920); Appendix A). More details were described in Table 3.

### 3.3. Serum or Plasma Iron Level and COVID-19

A total of seven studies included analyses of serum iron levels and the mortality of SARS-CoV-2 patients, involving 456 non-survivors and 1508 survivors. The serum iron level in dead patients was significantly lower than that in alive patients (SMD (95%CI): −0.483 (−0.597, −0.368), Z = 8.27, *p* for Z < 0.001; *I*^2^ = 0.9%, *p* for *I*^2^ =0.423; Table 2, Figure 4).

Seven studies were included in the serum iron and COVID-19 risk meta-analysis, including 492 cases and 704 controls. Figure 5 showed that lower serum iron levels were found in cases than controls (SMD (95%CI): −1.384 (−2.175, −0.592); Z = 3.43 *p* for Z = 0.001; *I*^2^ = 96.7%, *p* for *I*^2^ < 0.001; Table 3). In the subgroup analysis by continent, the same difference was discovered between Asia (SMD (95%CI): −3.403 (−5.974, −0.832); Appendix A) and Europe (SMD (95%CI): −0.580 (−0.791, −0.370); Appendix A). More details are described in Table 3.

A total of ten studies were included in the analysis of the relationship between the severity of COVID-19 and serum iron, involving 298 severe, 113 mild and 340 non-severe patients. Severe groups had lower iron levels than the mild groups (SMD (95%CI): −0.293 (−0.561, −0.024); Z = 2.13 *p* for Z = 0.033; *I*^2^ = 0.00%, *p* for *I*^2^ =0.545; Table 3, Appendix A) and non-severe groups (SMD (95%CI): −1.144 (−2.060, −0.227); Z = 2.45 *p* for Z = 0.014; *I*^2^ = 94.2%, *p* for *I*^2^ < 0.001; Table 3, Appendix A).

### 3.4. Hemoglobin Level and COVID-19

Regarding the connection between hemoglobin level and the mortality of COVID-19, a total of six studies were included, involving 276 the deceased and 1029 survivors. However, no significant differences were found between the hemoglobin levels of dead and recovered patients (SMD (95%CI): −0.186 (−0.571, 0.198), Z = 0.95, *p* for Z = 0.343; *I*^2^ = 82.5%, *p* for *I*^2^ < 0.001; Table 2, Figure 6). In addition, the subgroup analysis based on publication year indicated lower hemoglobin levels in the dead in 2021 (SMD (95%CI): −0.632 (−1.070, −0.194); Appendix A). More details about subgroup analysis are summarized in Table 2.

Seven studies were included in the hemoglobin level and COVID-19 risk meta-analysis, involving 367 cases and 511 controls. Figure 7 showed that lower hemoglobin levels were found in cases than controls (SMD (95%CI): −0.612 (−0.159, −0.065); Z = 2.19 *p* for Z = 0.028; *I*^2^ = 87.9%, *p* for *I*^2^ < 0.001; Table 3).

In terms of the analysis of severity, a total of twenty studies were included, including 1060 severe, 524 mild and 851 non-severe individuals. Though the difference between severe and mild groups (SMD (95%CI): −0.073 (−0.209, 0.064); Z = 1.040 *p* for Z = 0.298; *I*^2^ = 5.80%, *p* for *I*^2^ =0.386; Table 3) was not significant, the results demonstrated lower hemoglobin levels were found in severe groups than that in non-severe groups (SMD (95%CI): −0.394 (−0.703, −0.086); Z = 2.500 *p* for Z = 0.012; *I*^2^ = 86.50%, *p* for *I*^2^ < 0.001; Table 3, Appendix A).

### 3.5. Hepcidin Level and COVID-19

A total of three studies on the relationships between hepcidin and the mortality of COVID-19 were included in this meta-analysis, including 66 non-survivors and 224 survivors. Dead patients had slightly higher hepcidin levels than recovered patients, but this was not significant (SMD (95%CI): 0.447 (−0.287, 1.182); Z = 1.190 *p* for Z = 0.232; *I*^2^ = 84.8%, *p* for *I*^2^ =0.001; Table 2, Figure 8).

The analysis of COVID-19 risk included four studies totally, with 177 cases and 261 controls. Though higher hepcidin level was found in cases than that in controls, the difference was not significant (SMD (95%CI): 0.750 (−0.805, 2.306); Z = 0.95 *p* for Z = 0.345; *I*^2^ = 96.40%, *p* for *I*^2^ < 0.001; Table 3, Figure 9).

### 3.6. TIBC, TSAT and the Mortality of COVID-19

Four studies about TIBC were included in the analyses of mortality, including 382 non-survivors and 719 survivors. Analyses about TSAT involved three studies, 281 non-survivors and 458 survivors totally. The TIBC level was significantly lower in the death group than in the survivors group (SMD (95%CI): −0.612 (−0.900, −0.324), Z = 4.16, *p* for Z < 0.001; *I*^2^ = 71%, *p* for *I*^2^ = 0.016; Table 2, Figure 10). In addition, no significant differences were found in all analyses of TSAT (SMD (95%CI): −0.112 (−0.455, 0.231), Z = 0.64, *p* for Z = 0.521; *I*^2^ = 59.6%, *p* for *I*^2^ = 0.084; Table 2, Figure 11).

### 3.7. Sources of Heterogeneity and Publication Bias

Strong evidence of heterogeneity among studies was documented for the relationships between these iron-related biomarkers and mortality, clinical severity or risk in SARS-CoV-2 patients. The heterogeneity for the between-study was explored through the univariate meta-regression with the covariates of continent, gender, study types, years, ages and numbers of samples in the analysis. Meta-regression indicated that 75.02% of the heterogeneity in the COVID-19 risk analysis in serum iron was explained by continent. In terms of ferritin, publication year and sample size of the death group explained 18.33% and 14.78% of the heterogeneity of the mortality analysis, respectively; study type contributed 21.04% of the heterogeneity of overall analysis; and publication year contributed 20.33% of the heterogeneity of the severe–mild comparison. In addition, publication year and continent explained 46.95% and 46.09% of the heterogeneity of the moderate–mild analysis in ferritin, respectively. Regarding the heterogeneity of hemoglobin and mortality analysis, publication year accounted for 69.79%.

Considering the potential small-study effects, Galbraith plot showed the contribution of results from the different relevant studies to the heterogeneity. Then the “leave- one-out” sensitivity analyses used *I*^2^ > 50% as the criterion to evaluate the robustness of conclusion of ferritin. Twelve studies [8,38,47,48,51,55,57,58,59,60,62] on mortality (Figure 12) and four [17,20,21,24] studies on risk (Figure 13) were found to be the main reasons for the high heterogeneity. After excluding these studies, low heterogeneity and robust results without the small-study effect were demonstrated for mortality (SMD = 0; Z = 7.47; *p* < 0.001; *I*^2^ = 44.2%) and risk (SMD = 0; Z = 7.39; *p* < 0.001; *I*^2^ = 44.8%) analyses of ferritin.

The results of sensitivity analysis indicated that no individual study had an excessive influence on the pooled measure for all comparisons (Appendix A). Egger’s test reported no publication bias in mortality analysis (*p* = 0.930) or risk analysis (*p* = 0.129) of serum iron. However, the risk (*p* = 0.009) and mortality analyses (*p* < 0.001) of ferritin had significant publication bias. In terms of hemoglobin, Egger’s test demonstrated no publication bias in the analyses of mortality (*p* = 0.700) and risk (*p* = 0.483). Publication bias in mortality analyses of TIBC (*p* = 0.077) and TSAT (*p* = 0.358) was also not found. Moreover, no publication bias was found in the mortality (*p* = 0.065) and risk analyses (*p* = 0.699) of hepcidin.

## 4. Discussion

Our meta-analysis was based on 72 articles, containing 148 studies, of which 85 were about ferritin, 22 were about serum iron, 27 were about hemoglobin, 7 were about hepcidin, 3 were about TSAT and 4 were about TIBC. Mortality analyses included 29 studies on ferritin, 8 on serum iron, 6 on hemoglobin, 3 on hepcidin, 4 on TIBC and 3 on TSAT. Our results indicated that the serum iron and TIBC levels of the deceased were significantly lower than those of the survivors, but higher ferritin levels were found in the deceased. In terms of hemoglobin, hepcidin and TSAT, we did not find an association with death outcome. The risk analyses included 17 articles on ferritin, 7 on serum iron, 7 on hemoglobin and 4 on hepcidin. It was shown that cases had lower serum iron and hemoglobin levels, but higher ferritin levels than controls. There were not significant results in hepcidin analyses. Regarding the analyses of clinical severity, 39 studies on ferritin, 14 on hemoglobin and 7 on serum iron were included. All of them were found to be related to the severity of COVID-19.

According to previous studies, excessive inflammation is a characteristic of COVID-19 [82]. Iron plays an essential role in this process [9]. Serum iron, ferritin, hemoglobin, hepcidin, TIBC and TSAT represent the iron levels in the body; however, some of them have other important physiological functions.

Hepcidin is a main regulatory factor of iron metabolism that is associated with body iron level [83,84]. However, we did not observe significant links between that and COVID-19 onset or progression because of the limited numbers of studies. Differently from us, Denggao Peng et al. [85] classified the groups based on the clinical findings and reported that the hepcidin levels of severe COVID-19 cases were higher than those of non-severe cases. It is common that hepcidin is upregulated after a viral infection, especially for the COVID-19 patients with inflammation [86]. Moreover, thanks to this special change, hepcidin binds to ferroprotein and accelerates its degradation, so that iron uptake decreases and the iron storage in macrophages increases [87], influencing SARS-COV-2.

Ferritin is also recognized as an acute phase reactant of inflammation, influenced by the presence of iron, hepcidin [46,50,88,89,90] and pro-inflammatory cytokines. Inflammation promotes ferritin synthesis and release in the liver [91,92]. Moreover, ferritin also is the storage form of iron in macrophages, explaining the decrease in serum iron [93,94,95]. Henry et al. [90] found the associations between ferritin and COVID-19 severity. Similarly, we further documented the differences between their associations with COVID-19 risk and mortality. Though serum iron is an important indicator of disease, it cannot accurately represent the iron level because of the various related forms in the human body. Serum iron is essential for both humans and viruses [96,97,98]. In order to deprive SARS-CoV-2 of iron and support immunity, macrophages intake more iron, and the intestinal tract absorbs less, leading to a decrease in serum iron [81,82]. In addition, a study by Ehsani [99] reported a structural similarity between the hepcidin protein and the spiked glycoprotein cytoplasmic tail of SARS-CoV-2. This indicates that SARS-CoV-2 can simulate hepcidin’s action, contributing to the decreased serum iron [100]. A decreased hemoglobin level is usually a symbol of anemia, caused by decreased serum iron. TSAT reflects serum iron availability and is frequently used in clinical practice to detect states of iron deficiency or iron overload [101,102].

In meta-analysis, between-study heterogeneity is common. Thus, exploring the sources of between-study heterogeneity is essential. We performed univariate meta-regression, with covariables such as continent, study type, publication year and sample size. The regression results explained part of the heterogeneity in our meta-analysis, but there was still some heterogeneity not being detected. In addition, the Galbraith analysis indicated that high heterogeneity in ferritin resulted from twelve mortality analysis studies and five risk analysis studies. After excluding these studies, low heterogeneity and robust results without small-study effect were documented. However, the final results of ferritin were not changed.

There were some strengths in our meta-analysis: First of all, as far as we know, our study has unified a large number of studies on the associations of iron-related biomarkers with risk, clinical severity and mortality in COVID-19 patients, avoiding inaccurate conclusions of individual studies. More importantly, the random effects were used to estimate the pooled SMD. Thus, it was still possible to draw convincing results though the inconsistent measurement conditions and units for iron-related biomarkers in different studies.

However, our study has several limitations. Firstly, high heterogeneity was found in almost all indicator analyses, but some of analytical results were not explained by meta-regression or subgroup analysis. We cannot get a more accurate evaluation for the sources of the heterogeneity due to the lack of corresponding study-level covariates in the reported articles. Moreover, the differences in iron-related biomarkers’ reference values between females and males illustrated the importance of gender for our study. However, we could not conduct meta-analysis including gender for the reason that there was not enough information about it.

## 5. Conclusions

Our meta-analysis showed that the levels of serum iron and TIBC in dead patients were significantly lower than in survivors, and the ferritin level was higher in death groups than in survivors, whereas the relationship between hemoglobin and mortality was not significant. Moreover, serum iron and hemoglobin levels were lower in cases and negatively correlated with the severity; on the contrary, ferritin level was higher in cases. In addition, no statistically significant results were found in the hepcidin and TSAT levels of the severity and mortality groups. That was possibly due to the limited number of studies.

In conclusion, we found that ferritin, serum iron, hemoglobin and TIBC levels are closely associated with the risk, severity or mortality of COVID-19. These results provide strong evidence for the applications of iron-related biomarkers in the prediction of the COVID-19 occurrence and development. Moreover, lower serum iron and hemoglobin levels could provide clues for explaining the deteriorated process of COVID-19. However, future studies are needed to further confirm these results in future research.

## Figures and Tables

**Figure 1 nutrients-14-03406-f001:**
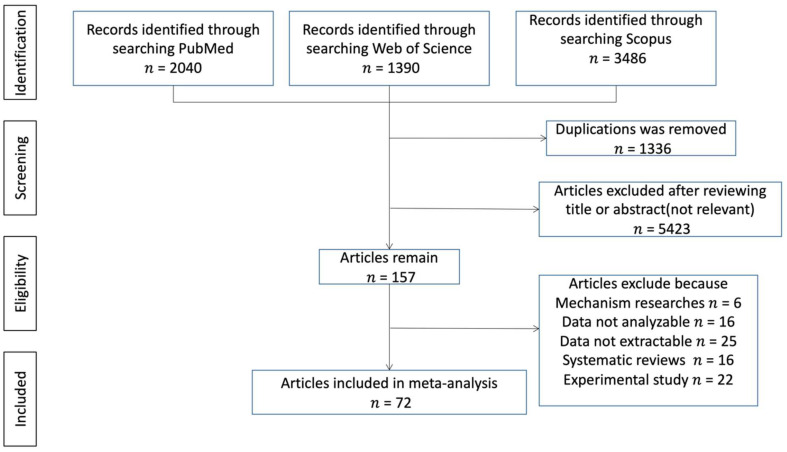
Flow diagram of the literature search.

**Figure 2 nutrients-14-03406-f002:**
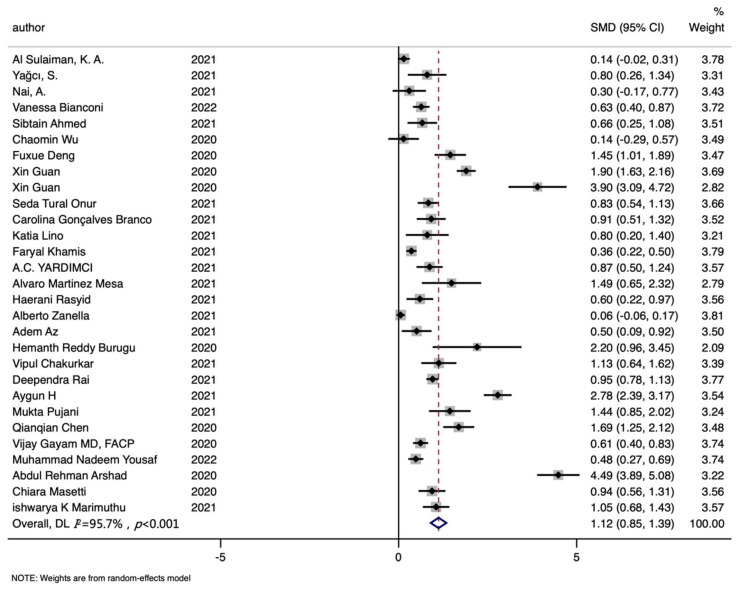
Forest plot of standard mean difference (SMD) with corresponding 95% confidence intervals (CIs) of studies [8,10,11,12,13,30,38,39,40,46,47,48,49,50,51,52,53,54,55,56,57,58,59,60,61,62,63,64] on ferritin levels in non-survivors and survivors. The solid diamond and horizontal line represent the study-specific effect and 95%CI, respectively; the size of the grey square is positively correlated with the weight distributed to each study in the meta-analysis. The center of open diamond with the vertical dashed line expresses the pooled SMD, and the width expresses the pooled 95%CI.

**Figure 3 nutrients-14-03406-f003:**
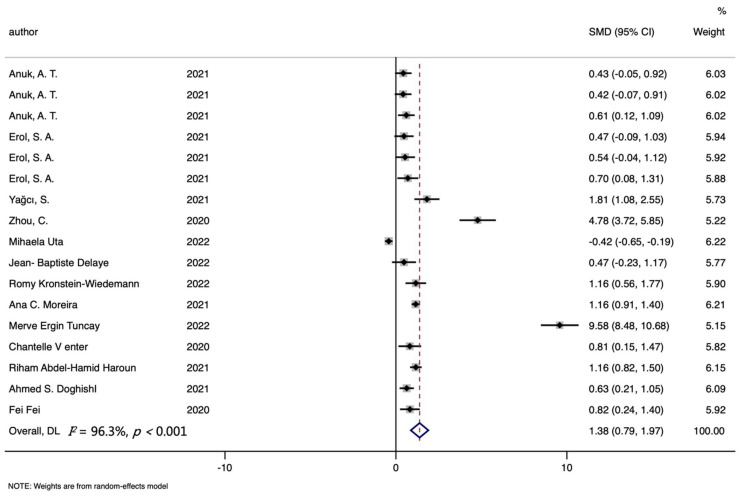
Forest plot of standard mean difference (SMD) with corresponding 95% confidence intervals (CI) of studies [10,17,18,19,20,21,24,36,37,42,43,44,72] on ferritin levels in COVID-19 cases and controls. The solid diamond and horizontal line represent the study-specific effect and 95%CI, respectively; the size of the grey square is positively correlated with the weight distributed to each study in the meta-analysis. The center of open diamond with the vertical dashed line expresses the pooled SMD, and the width expresses the pooled 95%CI.

**Figure 4 nutrients-14-03406-f004:**
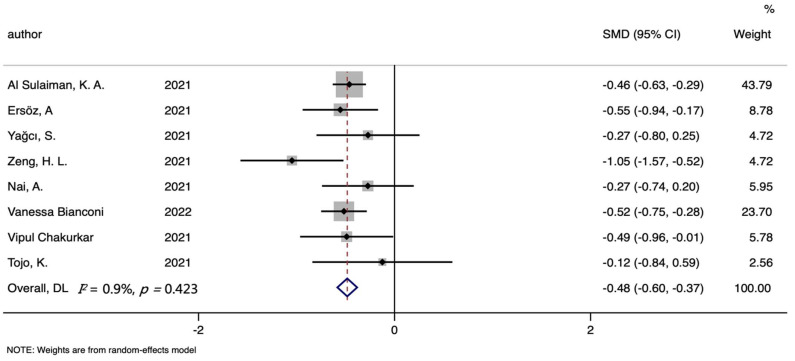
Forest plot of standard mean difference (SMD) with corresponding 95% confidence intervals (CIs) of studies [7,8,9,10,11,12,13,22] on serum iron levels in non-survivors and survivors. The solid diamond and horizontal line represent the study-specific effect and 95%CI, respectively; the size of the grey square is positively correlated with the weight distributed to each study in the meta-analysis. The center of open diamond with the vertical dashed line expresses the pooled SMD, and the width expresses the pooled 95%CI.

**Figure 5 nutrients-14-03406-f005:**
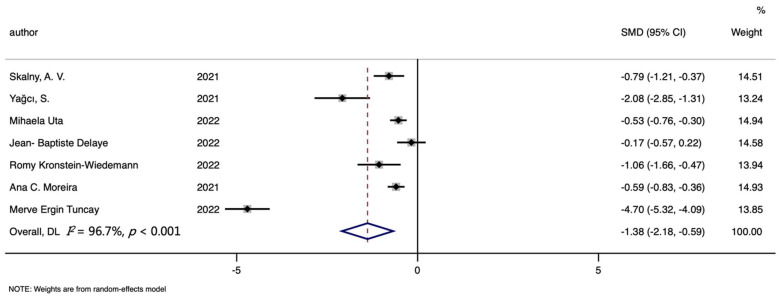
Forest plot of standard mean difference (SMD) with corresponding 95% confidence intervals (CIs) of studies [10,16,17,18,19,20,21] on serum iron levels in COVID-19 cases and controls. The solid diamond and horizontal line represent the study-specific effect and 95%CI, respectively; the size of the grey square is positively correlated with the weight distributed to each study in the meta-analysis. The center of open diamond with the vertical dashed line expresses the pooled SMD, and the width expresses the pooled 95%CI.

**Figure 6 nutrients-14-03406-f006:**
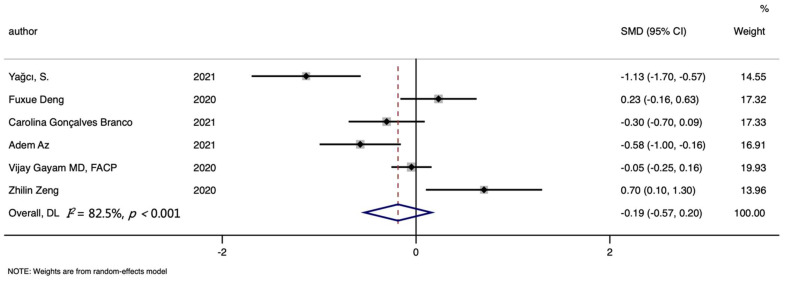
Forest plot of standard mean difference (SMD) with corresponding 95% confidence intervals (CIs) of studies [10,30,38,39,40,41] on hemoglobin levels in non-survivors and survivors. The solid diamond and horizontal line represent the study-specific effect and 95%CI, respectively; the size of the grey square is positively correlated with the weight distributed to each study in the meta-analysis. The center of open diamond with the vertical dashed line expresses the pooled SMD, and the width expresses the pooled 95%CI.

**Figure 7 nutrients-14-03406-f007:**
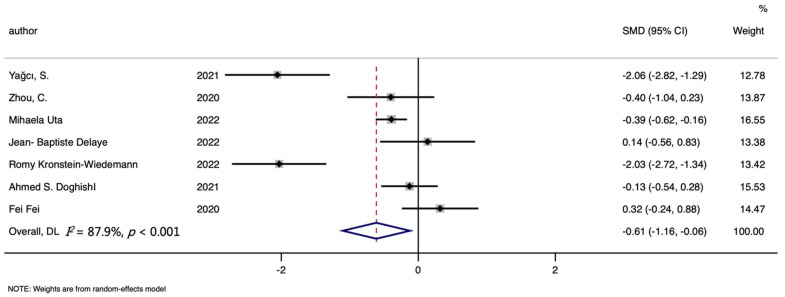
Forest plot of standard mean difference (SMD) with corresponding 95% confidence intervals (CIs) of studies [10,17,18,19,24,36,37] on hemoglobin levels in COVID-19 cases and controls. The solid diamond and horizontal line represent the study-specific effect and 95%CI, respectively; the size of the grey square is positively correlated with the weight distributed to each study in the meta-analysis. The center of open diamond with the vertical dashed line expresses the pooled SMD, and the width expresses the pooled 95%CI.

**Figure 8 nutrients-14-03406-f008:**
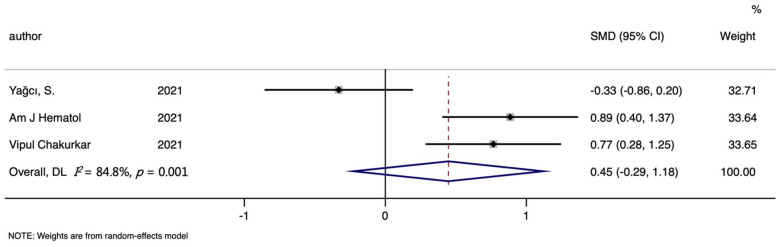
Forest plot of standard mean difference (SMD) with corresponding 95% confidence intervals (CIs) of studies [10,11,13] on hepcidin levels in non-survivors and survivors. The solid diamond and horizontal line represent the study-specific effect and 95%CI, respectively; the size of the grey square is positively correlated with the weight distributed to each study in the meta-analysis. The center of open diamond with the vertical dashed line expresses the pooled SMD, and the width expresses the pooled 95%CI.

**Figure 9 nutrients-14-03406-f009:**
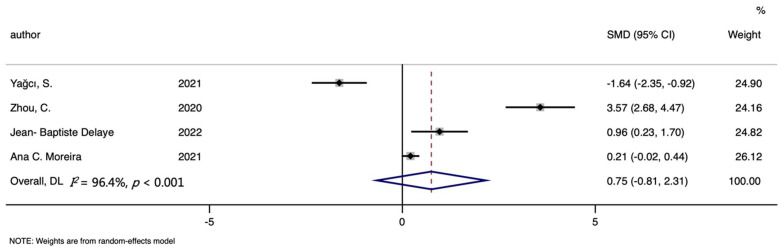
Forest plot of standard mean difference (SMD) with corresponding 95% confidence intervals (CIs) of studies [10,18,20,24] on hepcidin levels in COVID-19 cases and controls. The solid diamond and horizontal line represent the study-specific effect and 95%CI, respectively; the size of the grey square is positively correlated with the weight distributed to each study in the meta-analysis. The center of open diamond with the vertical dashed line expresses the pooled SMD, and the width expresses the pooled 95%CI.

**Figure 10 nutrients-14-03406-f010:**
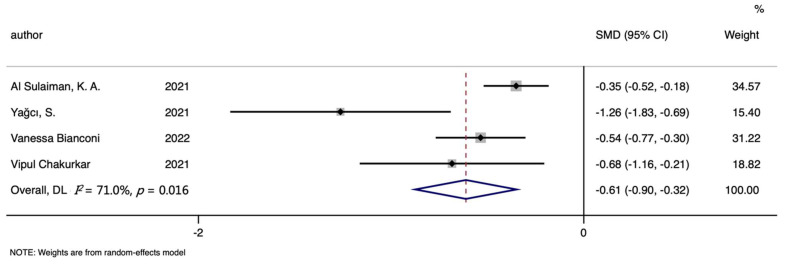
Forest plot of standard mean difference (SMD) with corresponding 95% confidence intervals (CIs) of studies [8,10,12,13] on TIBC levels in non-survivors and survivors. The solid diamond and horizontal line represent the study-specific effect and 95%CI, respectively; the size of the grey square is positively correlated with the weight distributed to each study in the meta-analysis. The center of open diamond with the vertical dashed line expresses the pooled SMD, and the width expresses the pooled 95%CI.

**Figure 11 nutrients-14-03406-f011:**
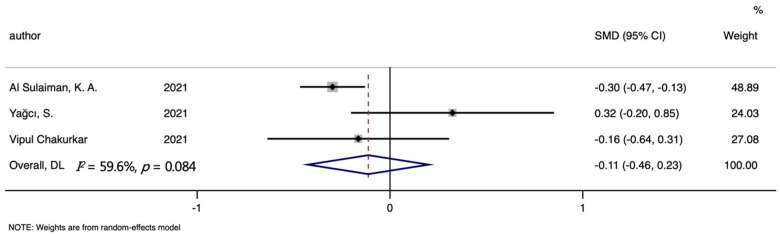
Forest plot of standard mean difference (SMD) with corresponding 95% confidence intervals (CIs) of studies [8,10,13] on TSAT levels in non-survivors and survivors. The solid diamond and horizontal line represent the study-specific effect and 95%CI, respectively; the size of the grey square is positively correlated with the weight distributed to each study in the meta-analysis. The center of open diamond with the vertical dashed line expresses the pooled SMD, and the width expresses the pooled 95%CI.

**Figure 12 nutrients-14-03406-f012:**
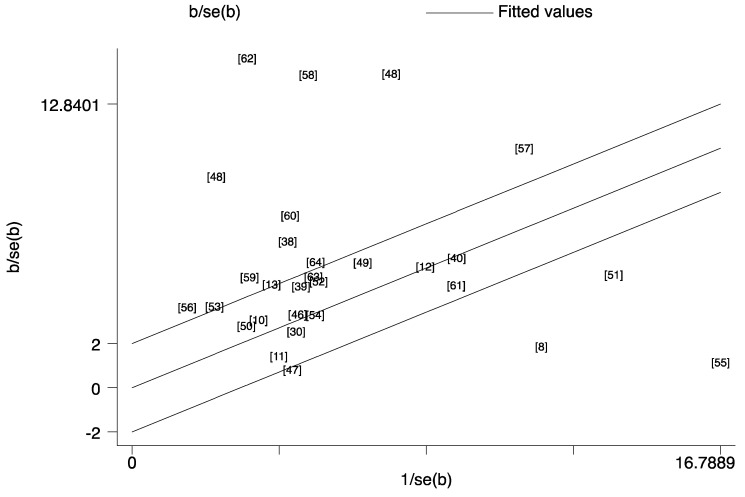
Galbraith plot for the contribution of results from the different studies [[8],[10],[11],[12],[13],[30],[38],[39],[40],[46],[47],[48],[49],[50],[51],[52],[53],[54],[55],[56],[57],[58],[59],[60],[61],[62],[63],[64],] about mortality for ferritin to the heterogeneity. This analysis was based on the relevant data listed in Table 1 and the number orders in the plot were same to the reference list.

**Figure 13 nutrients-14-03406-f013:**
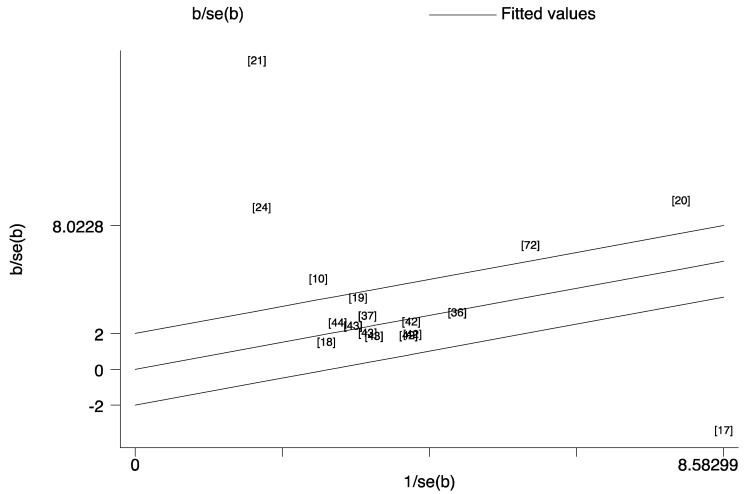
Galbraith plot for the contribution of results from the different studies [10,17,18,19,20,21,24,36,37,42,43,44,72] about risk for ferritin to the heterogeneity. This analysis was based on the relevant data listed in Table 1 and the number orders in the plot were same to the reference list.

**Table 1 nutrients-14-03406-t001:** Characteristics of 72 included studies of iron-related biomarkers.

Author (Year)	Country (Continent)	Study Type	Indicators	Group (G1/G2)	The Number of Samples (Group1/Group2)	Age Mean /Range (G1)	Mean (G1)	SD (G1)	Age Mean/Range (G2)	Mean (G2)	SD (G2)	Data Unit	Quality Assessment
Al Sulaiman, K.A. (2021) [8]	Saudi Arabia (AS)	C	Serum iron	dead/discharge	237/323	60	5.8	4.5	60	8.3	5.9	umol/L	6
			TIBC	dead/discharge	237/323	60	27.3	9.6	60	30.6	9.2	umol/L	
			TSAT	dead/discharge	237/323	60	18.6	13.8	60	23.5	18.5	%	
			Ferritin	dead/discharge	237/323	60	1247.3	1368.3	60	1069.2	1128.9	ug/L	
Allard, L. (2020) [65]	France (EU)	CS	Ferritin	Severe/no severe	34/74	58.9	1846.0	3328.0	68	676.0	696.0	µg/L	6
Anuk, A.T. (2021) [42]	Turkey (AS)	CC	Ferritin	1st trimester case/control	34/33	26	39.4	50.3	27	22.8	20.2	ng/mL	7
				2nd trimester case/control	33/32	27	61.0	156.1	28	14.1	9.2	ng/mL	
				3rd Trimester case/control	33/35	26	31.4	39.0	29	13.5	16.2	ng/mL	
Beigmohammadi, M.T. (2021) [77]	Iran (AS)	CS	Serum iron	severe/no-severe	20/40	56	38.0	27.0	50	52.0	38.8	mcg/dL	6
Dahan, S. (2020) [4]	Israel (AS)	CS	Ferritin	severe/no-severe	10/29	52.5	2817.6	3457.9	52.5	708.6	1074.5	ng/mL	6
				severe/mild	10/20	52.5	2817.6	3457.9	52.5	327.7	401.2	ng/mL	
				moderate/mild	9/20	52.5	1555.0	1578.1	52.5	327.7	401.2	ng/mL	
Erol, S.A. (2021) [43]	Turkey (AS)	CC	Ferritin	1st trimester case/control	24/26	26.3	47.0	59.3	26.34	26.2	22.8	ng/mL	7
				2nd trimester case/control	26/22	29.76	35.5	55.1	28.04	13.4	10.6	ng/mL	
				3rd Trimester case/control	21/22	26.895	31.6	45.4	26.3	9.4	5.6	ng/mL	
Ersöz, A. (2021) [9]	Turkey (AS)	CC	Serum iron	dead/discharge	29/281	69.2	25.3	22.7	55.8	42.1	31.0	mg/dL	6
Lv, Y. (2021) [6]	China (AS)	C	Ferritin	severe/no-severe	60/98	63.4	1088.7	285.6	63.4	328.1	216.2	ng/mL	7
			Serum iron	severe/no-severe	60/98	63.4	11.3	2.4	63.4	15.5	1.7	umol/L	
Skalny, A.V. (2021) [16]	Russia (EU)	CC	Serum iron	severe/control	50/44	NA	1.3	0.7	NA	1.9	0.7	ug/mL	7
Tojo, K. (2021) [7]	Japan (AS)	CC	Serum iron	dead/discharge	8/128	NA	38.5	41.1	NA	42.3	30.0	mg/dL	5
Yağcı, S. (2021) [10]	Turkey (AS)	CC	Serum iron	dead/discharge	23/36	63.63	309.0	200.9	63.63	369.3	235.3	ug/L	7
				critical/control	22/19	63.5	335.4	206.7	65.6	784.1	225.6	ug/L	
			TIBC	dead/discharge	23/36	63.63	1666.2	651.5	63.63	2388.4	515.7	ug/L	
			TSAT	dead/discharge	23/36	63.63	19.6	12.8	63.63	16.1	9.7	%	
			Hemoglobin	dead/discharge	23/36	63.63	103.8	26.3	63.63	129.3	19.7	ug/mL	
				critical/control	22/19	63.5	98.5	23.9	65.6	138.2	11.9	g/L	
			Ferritin	dead/discharge	23/36	63.63	1183.0	846.6	63.63	592.9	658.9	ug/L	
				critical/control	22/19	63.5	1205.8	853.5	65.6	68.8	43.6	ug/L	
			Hepcidin	dead/discharge	23/36	63.63	728.4	407.2	63.63	880.9	493.5	pg/mL	
				critical/control	22/19	63.5	603.3	244.4	65.6	992.8	230.7	pg/mL	
Yasui, Y. (2020) [70]	Japan (AS)	C	Ferritin	severe/mild	7/22	54.3	956.0	689.0	62.7	458.0	399.0	ng/mL	7
Zeng, H.L. (2021) [22]	China (AS)	C	Serum iron	dead/discharge	15/291	63	358.6	79.5	63	378.1	7.4	mg/L	6
				Severe/no severe	104/202	69	376.5	73.8	58	436.6	64.6	mg/L	
			Hemoglobin	severe/no severe	104/202	69	107.9	19.2	58	125.1	16.4	g/L	
			Ferritin	severe/no severe	104/202	69	683.7	556.5	58	338.3	259.3	ug/L	
Zhou, C. (2020) [24]	China (AS)	CC	Hemoglobin	severe/control	12/50	48.2	134.2	22.1	46.5	143.1	22.2	g/L	7
			Ferritin	severe/control	12/50	48.2	207.8	45.2	46.5	85.2	18.6	ng/mL	
			Hepcidin	severe/control	12/50	48.2	31.7	8.9	46.5	15.7	2.6	ng/mL	
Nai, A. (2021) [11]	Italy (EU)	C	Serum iron	dead/discharge	22/89	57.5	26.5	9.3	57.5	29.7	12.1	ug/dL	7
			Ferritin	dead/discharge	22/89	57.5	1939.1	1888.9	57.5	1494.9	1377.3	ug/L	
			Hepcidin	dead/discharge	22/89	57.5	516.6	236.8	57.5	329.2	205.3	ng/mL	
Uta, M. (2022) [17]	Romania (EU)	CC	Serum iron	case/control	95/351	NA	7.6	2.1	NA	8.8	2.3	umol/L	7
			Hemoglobin	case/control	95/351	NA	10.1	2.9	NA	11.0	2.1	g/dL	
			Ferritin	case/control	95/351	NA	21.4	4.2	NA	23.3	4.6	ng/mL	
Bianconi, V. (2022) [12]	Italy (EU)	CS	Serum iron	dead/discharge	101/261	79	26.7	15.0	72	38.9	26.1	μg/dL	6
			TIBC	dead/discharge	101/261	79	189.7	47.4	72	215.8	49.2	μg/dL	
			Ferritin	dead/discharge	101/261	79	822.7	798.7	72	462.0	451.0	ng/mL	
Delaye, J.B. (2022) [18]	France (EU)	CC	Serum iron	case/control	55/45	72.8	8.4	6.5	75.5	9.3	2.7	µmol/L	8
			Hemoglobin	case/control	55/45	72.8	115.9	22.2	75.5	113.1	18.7	g/L	
			Ferritin	case/control	55/45	72.8	829.3	940.4	75.5	485.5	425.9	µg/L	
			Hepcidin	case/control	55/45	72.8	43.7	43.8	75.5	12.9	10.6	nmol/L	
Zhao, K. (2020) [23]	China (AS)	CS	Serum iron	severe/mild	18/19	54.6	5.7	3.3	50.1	7.7	4.4	µmol/L	6
			Hemoglobin	critical/mild	13/19	65.3	135.1	24.1	50.1	128.1	26.4	g/L	
Kronstein-Wiedemann, R. (2022) [19]	Germany (EU)	CC	Serum iron	case/control	27/23	NA	5.2	3.2	NA	8.6	3.1	μmol/L	7
			Hemoglobin	case/control	27/23	NA	7.0	1.2	NA	9.0	0.7	mmol/L	
			Ferritin	case/control	27/23	NA	1082.7	1138.6	NA	103.2	119.5	µg/L	
Catherine, C. (2021) [14]	France (EU)	C	Serum iron	severe/mild	35/38	NA	6.4	6.6	NA	7.2	5.4	μM/L	6
			Ferritin	severe/mild (man)	22/20	NA	1431.0	44.0	NA	541.0	10.7	pg/dL	
			Ferritin	severe/mild (woman)	13/18	NA	1921.0	57.3	NA	334.0	11.3	pg/dL	
Moreira, A.C. (2021) [20]	Portugal (EU)	CC	Serum iron	case/control	127/176	≥18	28.2	18.7	≥18	47.7	40.0	µg/dL	6
			Ferritin	case/control	127/176	≥18	752.8	655.3	≥18	233.0	195.4	ng/mL	
			Hepcidin	case/control	127/176	≥18	69.8	53.3	≥18	54.5	83.9	nM	
Ahmed, S. (2021) [46]	Pakistan (AS)	CS	Ferritin	severe/no-severe	86/71	60.6	884.1	716.9	53.6	561.9	672.7	ng/mL	6
				dead/discharge	28/129	66.4	1107.6	784.9	55.5	650.2	667.1	ng/mL	
Kilercik, M. (2022) [15]	Turkey (AS)	C	Serum iron	Severe to Critical/Mild to Moderate	19/35	51.8	25.4	28.0	52	27.8	22.8	μmol/L	7
			Hemoglobin	Severe to Critical/Mild to Moderate	19/35	51.8	12.6	2.3	52	13.7	1.3	mmol/L	
			Ferritin	Severe to Critical/Mild to Moderate	19/35	51.8	740.1	831.8	52	357.4	345.5	mg/L	
Ergin Tuncay, M. (2022) [21]	Turkey (AS)	CC	Serum iron	case/control	116/46	60.8	27.9	7.9	37.5	69.2	10.7	µg/dL	7
			Ferritin	case/control	116/46	60.8	389.0	41.6	37.5	48.0	9.6	µg/L	
Wu, C. (2020) [47]	China (AS)	C	Ferritin	dead/discharge	44/40	68.5	1226.3	1104.0	50	1063.4	1259.7	ng/mL	6
Deng, F. (2020) [38]	China (AS)	CS	Hemoglobin	dead/discharge	50/50	68.7	128.2	18.7	62.4	123.5	21.4	g/L	6
			Ferritin	dead/discharge	50/50	68.7	1743.6	994.8	62.4	602.0	498.3	μg/L	
Guan, X. (2020) [48]	China (AS)	C	Ferritin	dead/discharge (1)	65/919	71	1568.9	1168.4	60.6	549.0	462.1	μg/L	6
				dead/discharge (2)	7/279	77.7	1793.6	1059.4	56.2	368.1	334.9	μg/L	
Tural Onur, S. (2021) [49]	Turkey (AS)	CS	Ferritin	dead/discharge	56/245	62	451.3	466.2	55	233.2	186.6	ng/mL	6
Branco, C.G. (2021) [39]	Portugal (EU)	CS	Hemoglobin	dead/discharge	34/96	83.1	3183.9	11.2	2.6	11.9	2.2	g/dL	6
			Ferritin	dead/discharge	34/96	83.1	3183.9	4248.3	70.6	958.7	1303.6	ug/dL	
Lino, K. (2021) [50]	Brazil (SA)	CS	Ferritin	dead/discharge	19/29	66.7	4207.7	3530.3	54.3	1717.7	2789.8	ng/mL	6
Khamis, F. (2021) [51]	Oman (AS)	CS	Ferritin	dead/discharge	257/745	63	1144.0	1060.9	51	820.8	837.8	ng/mL	6
Bozkurt, F.T. (2021) [78]	Turkey (AS)	CS	Ferritin	severe to critical/mild to moderate	23/70	61.9	670.4	631.3	37.2	108.4	95.1	ng/mL	7
García-Gasalla, M. (2021) [71]	Spain (EU)	CS	Ferritin	several/mild	32/49	61.7	454.8	383.4	53.5	164.8	176.4	ng/mL	6
Venter, C. (2020) [44]	South Africa (AF)	CC	Ferritin	case/control	33/13	53.1	394.8	416.1	55.6	105.5	77.7	ng/mL	8
Rahman, M.A. (2021) [25]	Bangladesh (AF)	C	Hemoglobin	severe/no-severe	108/198	54.1	12.6	6.0	45.6	12.7	1.4	g/dL	7
			Ferritin	severe/no-severe	108/198	54.1	651.9	793.3	45.6	86.4	34.8	ng/mL	
Yardımcı, A.C. (2021) [52]	Italy (EU)	CS	Ferritin	dead/discharge	30/692	57.2	575.4	419.6	57.2	316.0	292.0	μg/L	7
Kirtana, J. (2020) [67]	India (AS)	CS	Ferritin	moderate/mild	3/47	37.4	306.2	158.5	37.4	73.3	71.0	ng/mL	6
Martinez Mesa, A. (2021) [53]	Spain (EU)	C	Ferritin	dead/discharge	7/53	NA	1666.0	1217.0	NA	779.0	476.0	ng/mL	8
Rasyid, H. (2021) [54]	Indonesia (AS)	C	Ferritin	dead/discharge	31/264	56.6	3264.6	2941.1	46.2	1511.4	2941.2	ng/mL	7
Sukrisman, L. (2021) [68]	Indonesia (AS)	CS	Ferritin	severe/mild	6/39	50.9	2402.4	3886.9	50.9	1277.6	1939.8	μg/mL	6
				moderate/mild	64/39	50.9	1898.0	2514.0	50.9	1277.6	1939.8	μg/mL	
Zanella, A. (2021) [55]	Italy (EU)	CS	Ferritin	dead/discharge	426/834	67.4	1579.0	897.1	60	1514.2	1225.3	ng/mL	7
Az, A. (2021) [30]	Turkey (AS)	CS	Hemoglobin	dead/discharge	23/517	48	12.7	2.3	48	13.7	1.7	g/dL	6
				critical/mild	30/221	63.4	12.7	1.8	41.1	13.8	1.9	g/dL	
			Ferritin	dead/discharge	23/517	48	281.4	336.8	48	189.2	173.2	μg/L	
				several/mild	290/221	52.8	303.7	333.9	41.1	104.0	107.8	µg/L	
Burugu, H.R. (2020) [56]	India (AS)	CS	Ferritin	dead/discharge	3/47	41.7	1410.0	370.7	41.7	478.8	424.7	ng/mL	6
Chakurkar, V. (2021) [13]	India (AS)	C	Serum iron	dead/discharge	21/99	NA	23.9	9.1	NA	38.1	31.6	μg/dL	8
				severe/mild	41/22	59	30.6	19.3	38.5	42.7	28.4	μg/dL	
			TIBC	dead/discharge	21/99	NA	221.0	50.5	NA	269.8	74.9	μg/dL	
			TSAT	dead/discharge	21/99	NA	13.1	9.1	NA	15.1	12.4	%	
				severe/mild	41/22	59	15.0	10.8	38.5	14.7	10.9	%	
			Ferritin	dead/discharge	21/99	NA	810.2	894.8	NA	289.5	304.6	ng/mL	
				severe/mild	41/22	59	584.3	527.2	38.5	169.0	201.2	ng/mL	
			Hepcidin	dead/discharge	21/99	NA	235.3	120.8	NA	126.2	146.6	ng/mL	
Rai, D. (2021) [57]	India (AS)	CS	Ferritin	dead/discharge	188/498	58.7	847.6	769.4	48	362.6	365.8	ng/mL	7
Aygun, H. (2021) [58]	Turkey (AS)	C	Ferritin	dead/discharge	41/290	56	880.1	535.6	56	182.2	178.5	ng/mL	6
San Segundo, D. (2021) [69]	Spain (EU)	C	Ferritin	moderate–severe/mild	82/73	71.3	636.8	687.5	61.1	309.8	376.3	ng/mL	8
Pujani, M. (2021) [59]	India (AS)	CS	Ferritin	dead/discharge	15/85	NA	569.1	320.9	NA	276.5	176.5	ng/mL	8
				several/mild	13/61	≥18	624.6	314.8	≥18	235.2	174.43	ng/mL	
Haroun, R.A. (2021) [72]	Egypt (AF)	CC	Ferritin	severe to critical/mild to moderate	52/98	50.4	494.1	261.0	48.36	213.9	135.2	ng/mL	7
				case/control	150/50	43.43	361.1	252.5	45.8	104.4	50.9	ng/mL	
Sukrisman, L. (2021) [79]	Japan (AS)	CS	Ferritin	severe/no-severe	8/33	55.3	772.4	271.4	46.8	513.8	612.8	ng/mL	6
Chen, Q. (2020) [60]	China (AS)	CS	Ferritin	dead/discharge	46/68	65.9	1315.1	653.2	58.6	423.8	423.8	ng/mL	6
Gayam, V. (2020) [40]	America (NA)	C	Hemoglobin	dead/discharge	132/276	71	12.6	2.4	63	12.7	1.9	g/dL	7
			Ferritin	dead/discharge	132/276	71	1221.1	859.0	63	795.5	596.9	ng/dL	
Ghweil, A.A. (2020) [31]	Egypt (AF)	C	Hemoglobin	severe to critical/mild to moderate	30/36	62.6	12.6	0.9	55.5	12.7	1.2	g/dL	7
			Ferritin	severe to critical/mild to moderate	30/36	62.6	440.3	87.3	55.5	268.6	57.5	ng/mL	
Ramadan, H.K. (2020) [32]	Egypt (AF)	CS	Hemoglobin	severe/mild	60/66	NA	12.7	1.8	NA	12.8	1.7	g/dL	6
			Ferritin	severe/mild	60/66	NA	638.4	62.4	NA	258.1	47.6	ng/mL	
			Ferritin	moderate–severe/mild	134/66	NA	471.5	35.9	NA	258.1	47.6	ng/mL	
Zeng, Z. (2020) [41]	China (AS)	CS	Hemoglobin	dead/discharge	14/54	60.9	140.0	8.6	60.9	127.6	19.3	g/L	6
Yamamoto, A. (2021) [33]	Japan (AS)	CS	Hemoglobin	severe/mild	9/63	69.2	14.2	1.1	42.8	14.2	2.0	g/dL	6
			Ferritin	moderate/mild	48/63	50.9	246.7	194.1	42.8	194.1	196.5	ng/mL	
Abdelhakam, D.A. (2021) [34]	Egypt (AF)	CS	Hemoglobin	severe/mild	66/58	49.6	14.1	1.4	43.5	13.8	2.7	g/dL	7
			Ferritin	severe/mild	66/58	49.6	821.8	583.6	43.5	213.9	123.9	ng/mL	
Yousaf, M.N. (2022) [61]	Pakistan (AS)	CS	Ferritin	dead/discharge	135/251	56.5	810.0	409.0	52.7	593.0	471.0	ng/mL	6
Emsen, A. (2021) [35]	Turkey (AS)	CC	Hemoglobin	severe/mild	15/26	49.7	14.1	1.6	44.4	13.5	1.8	g/L	8
			Ferritin	severe/mild	15/26	49.7	294.7	255.9	44.4	137.6	255.4	ng/mL	
Doghish, A.S. (2021) [36]	Egypt (AF)	CC	Hemoglobin	case/control	171/26	41.4	13.3	1.9	42.9	13.6	2.3	g/dL	8
			Ferritin	case/control	171/26	41.4	349.1	403.7	42.9	110.8	91.6	ng/mL	
Fei, F. (2020) [37]	England (EU)	CC	Hemoglobin	case/control	24/26	65.4	12.4	5.3	56.3	11.1	2.7	g/dL	7
			Ferritin	case/control	24/26	65.4	1294.8	1624.0	56.3	354.8	328.0	ng/mL	
Bats, M.L. (2021) [66]	France (EU)	C	Ferritin	severe/no-severe	97/106	67.2	1228.2	1233.4	59.7	417.3	405.1	ng/mL	6
Arshad, A.R. (2020) [62]	Pakistan (AS)	CS	Ferritin	dead/discharge	22/216	41.2	1680.6	714.6	41.2	256.5	246.5	ng/mL	6
				several/mild	45/157	41.2	1267.0	998.2	41.2	183.1	135.1	ng/mL	
				moderate/mild	36/157	41.2	527.0	444.5	41.2	183.1	135.1	ng/mL	
Aly, M.M. (2021) [26]	Egypt (AF)	CS	Hemoglobin	severe/no-severe	165/185	54.6	11.6	2.2	38.1	12.4	2.2	g/dL	7
			Ferritin	severe/no-severe	165/185	54.6	249.3	263.3	38.1	223.4	254.0	mcg/mL	
Garcia-Gasalla, M. (2022) [80]	Spain (EU)	CS	Ferritin	severe to critical/mild to moderate	81/58	56	820.4	675.5	49.2	279.2	341.3	ng/mL	6
Huang, H. (2021) [27]	China (AS)	CS	Hemoglobin	severe/no-severe	21/43	61.4	128.2	16.3	41.2	138.1	13.8	g/L	7
			Ferritin	severe/no-severe	21/43	61.4	766.1	564.4	41.2	304.3	251.9	ng/mL	
Masetti, C. (2020) [63]	Italy (EU)	C	Ferritin	dead/discharge	33/196	75.2	1332.0	1675.0	58.3	577.0	545.0	ng/mL	7
Nizami, D.J. (2021) [81]	UAE (AS)	CS	Ferritin	severe/no-severe	18/75	≥18	4169.8	4954.6	≥18	381.3	3.5	ng/mL	6
Sana, A. (2022) [28]	India (AS)	CS	Hemoglobin	severe/no-severe	69/81	≥18	13.9	1.5	≥18	14.4	1.8	g/dL	6
			Ferritin	severe/no-severe	69/81	≥18	596.0	661.7	≥18	419.6	408.1	ng/mL	
Huang, C.Y. (2022) [29]	China (AS)	CS	Hemoglobin	severe/no-severe	86/142	66.2	13.6	1.8	55.8	13.8	1.8	g/dL	7
			Ferritin	severe/no-severe	86/142	66.2	1200.6	897.3	55.8	571.9	516.8	ng/mL	
Marimuthu, A.K. (2021) [64]	India (AS)	CS	Ferritin	dead/discharge	35/186	60	902.7	851.1	60	403.7	364.6	ng/mL	6

The details of the quality assessment are in Appendix A. SA: South America. AF: Africa. AS: Asia. EU: Europe. CC: case-control study. CS: cross-sectional study. C: cohort. SD: standard deviation. G: group. NA: Not available. The quality of studies was assessed by the Newcastle–Ottawa quality assessment scale.

**Table 2 nutrients-14-03406-t002:** Subgroup analyses of studies on the associations of iron-related biomarkers with mortality in SARS-CoV-2 patients.

Subgroups	N of Studies	SMD (95%CI)	Test of SMD = 0		Heterogeneity	
			Z	*p* for Z	*I* ^2^	*p* for *I*^2^
Serum iron	8	−0.483 (−0.597, −0.368)	8.27	<0.001	0.90%	0.423
Ferritin	29	1.121 (0.854, 1.388)	8.22	<0.001	95.70%	<0.001
Year						
2020	9	1.881 (1.137, 2.625)	4.95	<0.001	96.70%	<0.001
2021	18	0.847 (0.575, 1.119)	6.1	<0.001	93.90%	<0.001
2022	2	0.550 (0.393, 0.707)	6.86	<0.001	0.00%	0.347
Hemoglobin	6	−0.186 (−0.571, 0.198)	0.950	0.343	82.50%	<0.001
Year						
2020	3	0.215 (−0.168, 0.598)	1.100	0.272	67.40%	0.047
2021	3	−0.632 (−1.070, −0.194)	2.830	0.005	64.40%	0.060
Hepcidin	3	0.447 (−0.287, 1.182)	1.190	0.232	84.80%	0.001
TSAT	3	−0.112 (−0.455, 0.231)	0.64	0.521	59.60%	0.084
TIBC	4	−0.612 (−0.900, −0.324)	4.16	<0.001	71.00%	0.016

**Table 3 nutrients-14-03406-t003:** Subgroup analyses of studies on the associations of iron-related biomarkers with severity in SARS-CoV-2 patients.

Subgroups	N of Studies	SMD (95%CI)	Test of SMD = 0		Heterogeneity	
			Z	*p* for Z	*I* ^2^	*p* for *I*^2^
Serum iron						
Overall	7	−1.384 (−2.175, −0.592)	3.43	0.001	96.70%	<0.001
Continent						
Asia	2	−3.403 (−5.974, −0.832)	2.59	0.009	96.30%	<0.001
Europe	5	−0.580 (−0.791, −0.370)	5.41	<0.001	48.60%	0.100
Severe-Mild	4	−0.293 (−0.561, −0.024)	2.13	0.033	0.00%	0.545
Severe-non-Severe	3	−1.144 (−2.060, −0.227)	2.45	0.014	94.20%	<0.001
Ferritin						
Overall	17	1.383 (0.792, 1.975)	4.58	<0.001	96.30%	<0.001
Design						
Cross-Section Study	3	3.935 (−0.739, 8.608)	1.65	0.099	99.00%	<0.001
Case-Control Study	14	0.872 (0.443, 1.300)	3.98	<0.001	92.40%	<0.001
Severe-non-Severe	13	0.864 (0.389, 1.338)	3.57	<0.001	96.00%	<0.001
Severe-Mild	18	1.414 (0.995, 1.834)	6.61	<0.001	92.70%	<0.001
Year						
2020	5	2.652 (1.035, 4.269)	3.21	0.001	96.50%	<0.001
2021	11	1.037 (0.735, 1.340)	6.72	<0.001	77.70%	<0.001
2022	2	0.885 (0.583, 1.188)	5.74	<0.001	0.00%	0.407
Moderate-Mild	8	1.551 (0.535, 2.566)	2.99	0.003	97.20%	<0.001
Year						
2020	4	2.802 (0.678, 4.925)	2.59	0.010	97.40%	<0.001
2021	4	0.400 (0.207, 0.592)	4.07	<0.001	0.00%	0.553
Continent						
Asia	6	0.976 (0.362, 1.591)	3.11	0.002	88.00%	<0.001
Europe	1	0.581 (0.259, 0.903)	3.54	<0.001	/	/
Africa	1	5.319 (4.718, 5.920)	17.34	<0.001	/	/
Hemoglobin						
Overall	7	−0.612 (−1.159, −0.065)	2.190	0.028	87.90%	<0.001
Severe-non-Severe	6	−0.394 (−0.703, −0.086)	2.500	0.012	86.50%	<0.001
Severe-Mild	8	−0.073 (−0.209, 0.064)	1.040	0.298	5.80%	0.386
Hepcidin						
Overall	4	0.750 (−0.805, 2.306)	0.95	0.345	96.40%	<0.001

## Data Availability

The data used in this review come from published articles, all of which are identified in the references. The data used in the meta-analysis have been provided in the tables.

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
