# Peer review of "The Associations of Iron Related Biomarkers with Risk, Clinical Severity and Mortality in SARS-CoV-2 Patients: A Meta-Analysis"

_nutrients, 2022, doi:10.3390/nu14163406_

Round 1

Reviewer 1 Report

Major comments:

 The paper is based on a valuable literature base. About 10 % of the 92 rich literature items are older than the last 4 years of the SARS-COV-2 pandemic.

There is an inaccuracy between data in Fig. 1 and the description: "After screening the duplicated and irrelative articles, 72 articles were included. Among them, 85 articles were further excluded because: 6 were mechanism research; 16 didn't analyze the data; 25 expressed the data in odds ratio, relative risk or correlation coefficient; 16 were systemic reviews; and 22 were experimental studies. Eventually, a total of 72 eligible articles were involved in this meta-analyses."

I think it probably should be: "After screening the duplicated and irrelative articles, 157 articles were included.

Instead of: "After screening the duplicated and irrelative articles, 72 articles were included.

Poor legibility of descriptions in Figures 5, 6, 7, 8, 9 AND FIG. 1 SUPLEMENT – section A - despite the use of maximum magnification and unacceptable - overlapping description in Fig. 12 and Fig. 13.

 Use please proper terms and abbreviations. Necessary explanation of abbreviation in the abstract - TIBC and ARDS and TSAT in further text. In the limitation of study - add information about different data for men and women for e.g. iron reference values - so it is useful to include gender of patients and controls.

 Minor comments:

 Instead of:

The Association of Iron Related Biomarkers with Risk, Clinical Severity and Mortality in SARA-COV-2 Patients: A meta-analysis

Should be:

 The Association of Iron Related Biomarkers with Risk, Clinical Severity and Mortality in SARS-COV-2 Patients: A meta-analysis

 Instead of:

Considering that individual study may not have the sufficient power to obtain a reliable conclusion, this meta-analysis was conducted to: (1) Summarize and evaluate the results of numerous papers on the relationships between serum iron, ferritin, hemoglobin, TIBC, TSAT, hepcidin levels and the mortality and clinical severity in SARA-COV-2 patients.

Should be:

Considering that individual study may not have the sufficient power to obtain a reliable conclusion, this meta-analysis was conducted to: (1) Summarize and evaluate the results of numerous papers on the relationships between serum iron, ferritin, hemoglobin, TIBC, TSAT, hepcidin levels and the mortality and clinical severity in SARS-COV-2 patients.

Instead of:

Our keyword combinations include (iron OR ferritin) AND (COVID-19 OR human coronavirus disease 2019 OR SARA-COV-2 OR severe acute respiratory syndrome coronavirus

Should be:

Our keyword combinations include (iron OR ferritin) AND (COVID-19 OR human coronavirus disease 2019 OR SARS-COV-2 OR severe acute respiratory syndrome coronavirus

Instead of:

A total of twenty-nine studies assessed the connections between ferritin levels and the mortality of SARA-COV-2 patients in this meta-analysis, involving 2131 non-survivors and 7813 survivors.

Should be:

A total of twenty-nine studies assessed the connections between ferritin levels and the mortality of SARA-COV-2 patients in this meta-analysis, involving 2131 non-survivors and 7813 survivors.

Instead of:

A total of seven studies included analyses of serum iron levels and the mortality of SARA-COV-2 patients, involving 456 non-survivors and 1508 survivors.

Should be:

A total of seven studies included analyses of serum iron levels and the mortality of SARA-COV-2 patients, involving 456 non-survivors and 1508 survivors.

Instead of:

Strong evidence of heterogeneity among studies was documented for the relationships between these iron related biomarkers and mortality, clinical severity or risk in SARA-COV-2 patients.

Should be:

Strong evidence of heterogeneity among studies was documented for the relationships between these iron related biomarkers and mortality, clinical severity or risk in SARS-COV-2 patients.

Instead of:

Moreover, thanks to this special change, hepcidin binds to ferroprotein and accelerates its degradation, so that iron uptake decreases and the iron storage in macrophages increases 77, influencing SARA- COV-2.

Should be:

Moreover, thanks to this special change, hepcidin binds to ferroprotein and accelerates its degradation, so that iron uptake decreases and the iron storage in macrophages increases 77, influencing SARS- COV-2.

Instead of:

In order to deprive SARA-COV-2 of iron and support immunity, macrophages will intake more iron and intestinal tract absorbs less, leading to the decrease of serum iron.81

Should be:

In order to deprive SARS-COV-2 of iron and support immunity, macrophages will intake more  iron and intestinal tract absorbs less, leading to the decrease of serum iron.81

Supplemental  TABLE 1

Instead of:

Huang, C.Y.(20220

Should be:

Huang, C.Y.(2022)

Instead of:

Supplemental Figure. 1 Forest plot of standard mean difference (SMD) with corresponding 95% confidence interval (CI) of studies on serum iron (A, B), ferritin (C, D, E), hemoglobin (F, G) levels about severity meta0-analysis.

Should be:

Supplemental Figure. 1 Forest plot of standard mean difference (SMD) with corresponding 95% confidence interval (CI) of studies on serum iron (A, B), ferritin (C, D, E), hemoglobin (F, G) levels about severity meta-analysis.

Reviewer 2 Report

The manuscript entitled ‘The Association of Iron Related Biomarkers with Risk, Clinical Severity and Mortality in SARA-COV-2 Patients: A meta-analysis’ presents interesting issue, however it must be corrected and resubmitted.

 -       Title: It should be ‘SARS-CoV-2 Patients’ instead of ‘SARA-COV-2 Patients’

-       Abstract: It should be ‘SARS-CoV-2’ instead of ‘SARSCoV-2’

-       There are no line numbers, I cannot write what lines have errors.

-       Instead of p=0.0000 authors should report p < 0.0001

-       All abbreviations when used for the first time must be explained

-       There are many grammatical errors, typos (e.g. ‘non-survuvors’)

-       If literature search was made of the wrong assumption misspelling of word SARS-COV-2 (authors used “SARA-COV-2”) - we cannot trust that all relevant articles will be found. Authors should conduct the search once again and resubmit the article.

Round 2

Reviewer 2 Report

The authors have made a effort in answering my comments. However, the major weaknesses of the research still persist in the current version. Authors did not respond if the literature search was conducted properly – if authors search “SARA-COV-2” or “SARS-COV-2” – the results may be differ. This aspects is crucial for faithfulness of the meta-analysis (all relevant publication must be induced)

Meta-analysis formal, epidemiological study design used to systematically assess the results of previous research with very a rigorous manner. In this manicurist there are lot of mistake (including misspelling of SARS-COV-2, so the data extraction could be biases).

There are serious problem with provided data – eg. (1) In table 1 publication by Tojo has ‘NA” for age, whereas in publication such data is presented in table 1. (2) Al Sulaiman, K. A.; Aljuhani, O.; Eljaaly, K.; Alharbi, A. A.; Al Shabasy, A. M.; Alsaeedi, A. S.; Al Mutairi, M.; Badreldin, H. A.; Al Harbi, S. A.; Al Haji, H. A.; Al Zumai, O. I.; Vishwakarma, R. K.; Alkatheri, A., Clinical features and outcomes of critically ill patients with coronavirus disease 2019 (COVID-19): A multicenter cohort study. Int J Infect Dis 2021, 105, 180-187 – I could not find any data that were presented in Table 1!

Table 1 – on the basis on what criteria “Quality Assessment” was conducted?

Table 1 – it will be more suitable if authors recalculated data for same unit for each indicators. It is very difficult to compare Serum iron in umol/l and mcg/dl

The table 1 should be divide into two tables to better readability.  

Figure 1 must be before table 1

There are misspelling on the Figure (e.g. systemic review?????).
